# Evidence and Uncertainties on Lipoprotein(a) as a Marker of Cardiovascular Health Risk in Children and Adolescents

**DOI:** 10.3390/biomedicines11061661

**Published:** 2023-06-08

**Authors:** Simonetta Genovesi, Marco Giussani, Giulia Lieti, Antonina Orlando, Ilenia Patti, Gianfranco Parati

**Affiliations:** 1School of Medicine and Surgery, Milano-Bicocca University, 20126 Milan, Italy; g.lieti@campus.unimib.it (G.L.); a.patti8@campus.unimib.it (I.P.); dir.sci@auxologico.it (G.P.); 2Istituto Auxologico Italiano, Istituto Ricovero Cura Carattere Scientifico (IRCCS), 20135 Milan, Italy; dottormarcogiussani@gmail.com (M.G.); a.orlando@auxologico.it (A.O.)

**Keywords:** lipoprotein(a), children, adolescents, cardiovascular risk

## Abstract

Lipoprotein(a) (Lp(a)) is made up of apoprotein(a) (apo(a)) and an LDL-like particle. The *LPA* gene encodes apo(a) and thus determines the characteristics and amount of apo(a) and Lp(a). The proportion of Lp(a) in each individual is genetically determined and is only minimally modifiable by the environment or diet. Lp(a) has important pro-atherosclerotic and pro-inflammatory effects. It has been hypothesized that Lp(a) also has pro-coagulant and antifibrinolytic actions. For these reasons, high Lp(a) values are an important independent risk factor for cardiovascular disease and calcific aortic valve stenosis. Numerous studies have been performed in adults about the pathophysiology and epidemiology of Lp(a) and research is under way for the development of drugs capable of reducing Lp(a) plasma values. Much less information is available regarding Lp(a) in children and adolescents. The present article reviews the evidence on this topic. The review addresses the issues of Lp(a) changes during growth, the correlation between Lp(a) values in children and those in their parents, and between Lp(a) levels in children, and the presence of cardiovascular disease in the family. Gaining information on these points is particularly important for deciding whether Lp(a) assay may be useful for defining the cardiovascular risk in children, in order to plan a prevention program early.

## 1. Introduction

Lipoprotein(a) (Lp(a)) was first described by K. Berg in 1963 [1]. For several years, interest in this lipoprotein was modest, until the sequencing of the *LPA* gene encoding apolipoprotein(a) (apo(a)), a key constituent of Lp(a) [2], and the subsequent recognition of Lp(a) as an independent risk factor for cardiovascular disease by epidemiological and Mendelian randomization studies [3,4]. Later, high Lp(a) values were also associated with calcific aortic valve stenosis [5] and increased risk of ischemic stroke [6,7].

## 2. Structure and Features of Lp(a)

Lp(a) consists of a very low-density lipoprotein (LDL)-like particle with apoprotein B100 being bound via a single disulfide link to a single apoprotein(a) (apo(a)) (Figure 1).

Apo(a) is a glycoprotein that has structural similarities with plasminogen. Plasminogen is made from a protein chain with a terminal part with protease activity joined to five aggregates of about 90 amino acids that have been named kringles because of their shape resembling that of a typical Northern European cake. In plasminogen, the kringles are present in single copy and are named with Roman numerals from KI to KV. Unlike plasminogen, apo(a) has an inactive protease domain and does not contain KI, KII, or KIII kringles, but consists of only one KV subtype and ten KIV subtypes, which are present in single copy except for the KIV_2_ subtype that can repeat from one to forty times [8] (Figure 2). Polypeptide structures similar to kringle domains are found in several enzymes (thrombin, tissue plasminogen activator, hepatocyte growth factor) as well as in plasminogen and Lp(a). The KVs of plasminogen and apo(a) have strong structural homology, as do the KIVs of plasminogen and Lp(a). However, KV and KIV are quite different from each other, both structurally and as amino acid composition. In contrast, the various subtypes 1 to 10 of KIV have important structural homologies and differ in their composition by only a few amino acids. These small differences may, however, account for different activities such as, for example, the ability to create disulfide bonds between KIV9 and apoB100. The specific function of KIV2 is not known, nor is the physiological function of Lp(a). The characteristic of KIV2 is its ability to be replicated a highly variable number of times in the structure of apo(a) [9].

Thus, the length of apo(a) is highly variable and its molecular weight has a range from 275 to 800 kDa. Apo(a) contains a fair amount of oligosaccharides, in particular, sialic acid and other N-glycans and O-glycans, which have structural and protective functions against the activity of proteolytic enzymes [10,11]. Another relevant aspect in the structure of apo(a) is the presence of oxidized phospholipids, mainly attached to the KIV_10_ subtype. Considering also the proportion contained in the LDL-like component, Lp(a) is the major carrier of oxidized phospholipids [12], so individuals with high Lp(a) values are at greater risk of coronary heart disease and stroke. Apo(a) is produced exclusively by the endoplasmic reticulum of the liver, where it also undergoes post-translational modifications with the creation of a disulfide bond in each kringle that assumes the characteristic shape. The formation process is completed in the Golgi apparatus where the lipoprotein undergoes glycosylation [13]. The question of whether the assembly of Lp(a) with apo(a) and apoprotein B-100 takes place in the hepatocyte, on the surface or even outside, is not settled yet. Some studies suggest that an initial non-covalent bond between apo(a) and apoprotein B100 could be formed inside the hepatocyte [10], while the final disulfide bond that completes the formation of Lp(a) would be created outside the hepatocyte, in the Disse spaces of the hepatic sinusoids [14,15]. A small proportion of apo(a) remains unaggregated and is eliminated with urine; no particular function has been attributed to it [16]. The amount of circulating Lp(a) depends on how much apo(a) is produced by the liver, which, in turn, depends on the activity of the *LPA* gene, which, being made up of two codominant alleles, encodes two isoforms of apo(a) that may differ in length. In each individual, therefore, there are two different Lp(a)s which may have different molecular weights. The variability in Lp(a) levels in the general population is extremely high, and its plasma concentration can range from 0.1 to 300 mg/dL. Hepatic production of Lp(a) is 80–90% genetically determined; therefore, it is little affected by environmental factors and tends to be stable in adults [17]. Therefore, in this age group, a single Lp(a) assay may be sufficient to know the related risk profile of the individual. However, this point is a matter of debate, as it is reported that Lp(a) levels would change over time [18]; moreover, this occurs in the presence of certain diseases. First, it should be noted that Lp(a) behaves as an acute phase protein, so it tends to increase in inflammatory states [19] and this should be taken into account when scheduling the measurement. In addition, nephropathies are also associated with significant increases in Lp(a) levels, which can be explained by different mechanisms depending on the type of kidney disease. Already in the early stages of renal failure, an increase in Lp(a) can be shown, with increases proportional to the decrease in glomerular filtrate. However, in the nephrotic syndrome, Lp(a) can increase up to fourfold due to an increment of its synthesis by the liver [20]. Chronic kidney disease patients have a significantly increased risk of cardiovascular disease, related to the level of Lp(a) [21]. The amount of circulating Lp(a) is inversely proportional to the length of the apo(a), thus to the number of KIV_2_ kringles. Therefore, individuals with longer apo(a) isoforms of Lp(a), thus with higher molecular weight, produce less Lp(a) than those with shorter, lower molecular weight isoforms. The lower amount of Lp(a) produced by individuals in which the higher molecular weight isoforms predominate could be due to the longer time required for its production [22] or to the greater molecular weight isoforms that have not assumed the correct conformation being translocated outside the endoplasmic reticulum, ubiquitinated and degraded [23]. In addition to length, other genetic factors may influence the increased or decreased production of Lp(a). In particular, several single nucleotide polymorphisms of the *LPA* gene are known to be associated with increased or decreased apo(a) and Lp(a) production [24,25]. Regarding cardiovascular risk, it is currently believed that it would be related to the amount of circulating Lp(a) and not to its qualitative characteristics. Thus, subjects with higher plasma concentrations of this lipoprotein, who are those with lower molecular weight isoforms of Lp(a), are most at risk [3]. Lp(a) is metabolized mainly by the liver and to a lesser extent by the kidney. Several receptors are called upon for Lp(a) uptake, such as the BI scavenger receptor, plasminogen receptors, and LDL receptors, but these aspects are not yet fully elucidated [26]. The variability of Lp(a) gives reason for the difficulties there are in developing precise laboratory methods for its assay. In particular, when polyclonal antibodies, also directed toward the repeated part of apo(a) (KIV_2_), are used, larger isoforms tend to be overestimated, while smaller ones may be underestimated [27]. These difficulties can be overcome by performing analyses with calibrators containing Lp(a) of different lengths or with monoclonal antibodies directed only toward the single-copy portion of apo(a). Lp(a) assays are often expressed in mg/dL, as if directly measuring the entire lipoprotein, and thus, not only its protein part but also its cholesterol content, cholesterol esters, phospholipids, triglycerides, and carbohydrates. This is not quite correct because the assays reflect the quantitative molar ratio between apo(a) and the antibodies that interact with it. Therefore, it would be more correct to express Lp(a) values in nmol/L [27]. Again because of the variability of Lp(a), there can be no precise conversion factor between mg/dL and nmol/L. However, from a practical point of view, a conversion factor between 2 and 2.5 is still accepted [28].

## 3. Lp(a) and Cardiovascular Risk

Lp(a) is associated with an increased risk of cardiovascular events because of some of its structural and functional characteristics that promote atherosclerotic plaque formation and growth. In addition, it has been suggested that Lp(a) promotes coagulation processes following plaque rupture. This, however, is currently only a hypothesis and needs validation. Lp(a) contains a proportion of cholesterol that corresponds to about 30–40% of its weight [29]. This amount of cholesterol is less than that found in LDL and is negligible when circulating Lp(a) is in low concentration. However, in individuals who have high Lp(a) values, its contribution to the amount of atherogenic cholesterol can be significant, mainly due to the ease with which Lp(a) crosses the endothelium and accumulates in the inner layer of the artery wall, where it is able to activate smooth muscle cell proliferation and migration and foam cell formation [13]. Interestingly, high levels of Lp(a) promote the earlier stages of atherosclerosis and not just the more advanced phases. Imaging studies have shown that Lp(a) promotes the development of arterial wall inflammation [30] and that in the presence of coronary artery disease, high Lp(a) levels are associated with an increase in the amount of calcium and necrotic core volume from the atherosclerotic plaque [31,32]. The processes promoting inflammation and calcium deposition seem mainly related to the presence of oxidized phospholipids [33,34], which would be responsible for the secretion of chemotactic substances and proinflammatory cytokines, upregulation of adhesion molecules, and trans-endothelial migration of monocytes [30,35]. Finally, the presence in apo(a) of a protease-like domain similar to that of plasminogen, but inactive, suggests that Lp(a) might also have a role in promoting coagulation processes and reducing fibrinolysis. The latter effect of Lp(a) is still under debate [36]. However, it seems that Lp(a) has no role in venous thromboembolism events [37]. In summary, Lp(a) would have pro-atherosclerotic, pro-inflammatory, and possibly pro-coagulant and antifibrinolytic effects. Regarding calcific aortic valve stenosis, numerous studies have shown micro- and macro-calcifications of the aortic valve in adults aged 45 to 55 years with elevated Lp(a) values [5,38,39]. It is estimated that individuals with higher Lp(a) percentiles are three times more likely to develop calcific aortic valve stenosis than those with lower Lp(a) percentiles [38]. Furthermore, high plasma Lp(a) levels correlate with more rapid progression of valve stenosis [12]. The etiopathogenetic mechanisms explaining the association between Lp(a) and calcific aortic valve stenosis have yet to be elucidated; however, oxidized phospholipids would again be called into play, which, in addition to activating inflammatory processes, would stimulate the activation of genes that regulate osteoblastic processes in the cells of the valve interstitium [12,35].

Some considerations should be added regarding the relationship between hypolipidemic drug therapy and the levels of Lp(a). Statins have been reported to be associated with a tendency to increase Lp(a) levels, whereas ezetimibe would have no effect [40,41].

In contrast, apheresis [42] and drugs such as niacin [43] and PCSK9 inhibitors [44] are able to reduce plasma Lp(a) concentrations. However, while it has been shown that the decrease in Lp(a) levels induced by apheresis is approximately 75% and is associated, in subjects with extremely high Lp(a) levels, with a significant reduction in cardiovascular risk [42], the same clinical outcome is not achieved with the administration of niacin or PCSK9 inhibitors [45,46], which induce reductions in Lp(a) values of 20–30% [43,44]. On the basis of the current data, a significant reduction in cardiovascular risk by lowering Lp(a) awaits further studies. Of note, niacin and PCSK9 inhibitors do not currently have a pediatric indication.

## 4. Lp(a) in Adults

The plasma concentration of Lp(a) is widely variable, and in all populations, there are individuals with very-high Lp(a) values. However, important differences can be found in the distribution of Lp(a) in different ethnic groups. In fact, average Lp(a) values increase from Chinese, to South Asians, to Whites to Blacks, with average concentrations of 16, 19, 31, and 75 nmol/L, respectively [47]. Hispanics have values intermediate between South Asians and Caucasian Whites [48]. In general then, Black individuals are those with higher Lp(a) values; in fact, the mean value of the plasma Lp(a) concentration among Blacks is in the fifth quintile of the White population distribution [49]. These differences among the different ethnic groups would be largely explained by the greater presence of isoforms with lower KIV_2_ kringle number in the Black, Caucasian, and Hispanic populations, respectively, although mutations in the *LPA* gene and other yet unknown factors would play a non-secondary role [50,51]. A recent large study confirms the inverse relationship between the distribution of Lp(a) values and its isoforms and the different distribution among different ethnic groups [52]. The study shows that Chinese and South Asian populations have the lowest average Lp(a) values and largest isoforms, while Africans and Arabs have the highest concentrations and smallest isoforms. Europeans, Latin Americans, and Southeast Asians rank in between for both parameters. The same study shows a correlation between the presence of Lp(a) values greater than 50 mg/dL with the risk of myocardial infarction with an odds ratio of 1.48 (95% CI: 1.32–1.67) in the entire population. The risk increased from Africans, to Arabs, Chinese, Europeans, Latin Americans to Southeast Asians who had the highest odds ratio [52]. This study suggests that ethnicity should also be taken into account when assessing the cardiovascular risk associated with Lp(a) levels, as similar Lp(a) values might be associated with different patterns of cardiovascular risk in different ethnicities. Lp(a) values could also differ according to gender. However, the reports on this issue are conflicting: some authors report that males have higher values than females [53], while for other authors, the opposite would be true [47]. In males, Lp(a) values would remain consistent throughout adult life [54], whereas in females, there would be an increase after menopause [55,56,57,58]. In adults, the correlation between Lp(a) levels and cardiovascular risk has a linear and continuous pattern. In general, individuals with Lp(a) values less than 30 mg/dL can be considered at low risk and those with values greater than 50 mg/dL at high risk. Concentrations between 30 and 50 mg/dL are considered borderline [27]. It should be emphasized, however, that in the assessment of an individual’s cardiovascular risk, Lp(a) values are only one aspect, albeit an important one, that must be considered in the totality of the other known risk factors. The most recent European guidelines suggest that Lp(a) should be measured at least once in a lifetime in adults and that this information should be included in the overall estimate of the risk of developing atherosclerotic-based disease [27].

## 5. Lp(a) in Children and Adolescents

The number of Lp(a) studies performed in pediatric populations is much smaller than those performed in adults. This is because pediatric research on cardiovascular risk factors is often ancillary to that in adulthood, but also because cardiovascular events are exceptional in very young people. However, four studies have been published that relate to increased Lp(a) values to the occurrence of arterial ischemic stroke in infants and children [59,60,61,62]. In addition, a meta-analysis reported an OR of 6.27 (95% CI, 4.52 to 8.69) for ischemic stroke in children with elevated Lp(a) compared to children with normal values [63]. However, arterial ischemic stroke is a very rare condition in childhood, so this finding refers to a restricted aspect of the relationship between elevated Lp(a) levels and cardiovascular disease in youth. Studies that relate to Lp(a) measurement in children and adolescents are in small numbers and involve two time periods. The oldest studies date back to the 1990s, and publications have resumed in more recent times when interest around Lp(a) has grown, partly because of the possibility of specific therapy that could reduce its values.

Thus, the main questions are (i) how does Lp(a) vary during childhood and (ii) is there a correlation between Lp(a) values in children, those of their parents, and the presence of markers of atherosclerosis in adolescents and/or cardiovascular disease in the family?

In the available studies, the Lp(a) assay is expressed with different units: in milligrams/liter, in milligrams/deciliter, or in nanomoles/liter. In reporting the results of the different studies, we have kept the data as expressed in the original articles.

### 5.1. Changes in Lp(a) Values in Childhood

As for this topic, the questions are (i) how do Lp(a) levels vary during childhood; (ii) when do they reach adult values; (iii) will a child with high Lp(a) be an adult with high Lp(a) values?

A first study carried out in 1983 showed that Lp(a) values in neonates are very low compared to those in adults [64]. A subsequent study confirmed that the Lp(a) values at birth are very low and increase significantly by the seventh day of life and are still increasing even at 180 days after birth [65]. A later study reported Lp(a) values in 232 infants including 123 of white ethnicity and 109 of black ethnicity. The birth value of Lp(a) was 4 mg/dL with no differences by ethnicity and gender. The same study showed that Lp(a) values increased gradually from birth, reaching adult values already by the second year of life [66]. Schumacher and Wood assayed Lp(a) in 123 term infants showing a mean concentration of 13.9 mg/L in the umbilical cord blood and 10.2 mg/L in the capillary blood [67]. Subsequently, the same group published a study conducted on 625 infants and 221 children in the first year of life. The median value of Lp(a) in children less than 1 year old was 37.0 mg/L regardless of gender (males 37.1, females 37.0 mg/L) [68]. Wilcken et al. calculated the 50th and 95th percentile values of Lp(a) on a sample of more than 1000 children in the third–fifth days of life, which were found to be 30 and 130 mg/L, respectively. Repeat assays at month 8 in a subgroup of the same sample showed that the values were doubled [69]. A Turkish study conducted in 430 children with dosing at 7, 13, 24, and 36 months showed mean Lp(a) values of 84, 156, 134, and 136 mg/L, respectively. The authors also found a strong correlation between the Lp(a) concentrations of the four measurements in the individual child [70]. Two recent studies have made important contributions regarding the evolution of Lp(a) values from infancy and adolescence. In the first, the authors assayed Lp(a) in the umbilical cord, at birth at 2 months and at 15 months obtaining values of 2.2, 2.4, 4.1, and 14.6 mg/dL, respectively, showing a strong correlation between umbilical cord blood and venous blood values at birth and a moderate correlation between values at 2 and 15 months [71]. In the second, a group of Dutch researchers published data collected on a sample of nearly 3000 children and adolescents referred to an outpatient clinic dedicated to the treatment of dyslipidemia. The population had high mean LDL cholesterol values (184 mg/dL), and in about two-thirds of the cases, a diagnosis of familial hypercholesterolemia had been made and confirmed on genetic analysis. Lp(a) assays, performed by two different analytical methods, showed mean values of 117 and 103 mg/L. Lp(a) values increased with age in both the group of untreated individuals and those taking hypolipidemic drugs. In individuals in whom multiple assays had been performed at different times, at least a 20% increase in Lp(a) levels was observed. Individuals who showed high values when they were children maintained high values as adults. Finally, the researchers showed that in all of the study population, Lp(a) values increased into adulthood, and individuals treated with statins tended to have a greater increase. Subjects taking the statin/ezetimibe combination reached a plateau at age 15 years [72]. These data would confirm what has already been observed in adults in whom statin treatment would induce an increase in Lp(a) values not shown in those taking ezetimibe [40,41]. Finally, a recent study conducted in 416 Korean children with a mean age of 11.1 years, in which the mean values were 21.5 nmol/L, should be noted. The prevalence of individuals with Lp(a) values > 100 nmol/L (equal to about 50 mg/dL) was 11.3%. No age- or gender-related differences were evident [73].

### 5.2. Correlation between Pediatric Lp(a) Values and Clinical Data

In his study, Wilcken et al. [69] showed a correlation between Lp(a) values in children in the third–fifth day of life and those of their parents, and this correlation was even clearer in measurements performed at eight months. The study by de Boer [72] confirmed a correlation between Lp(a) values from eight years of age to adulthood, with an increase in values of about 20 percent. Qayum et al. assayed Lp(a) in 257 14-year-old adolescents referred to an outpatient cardiovascular prevention clinic based on the presence of dyslipidemia or early cardiovascular events in the family. The sample was divided into two groups based on the presence of Lp(a) values greater or less than 30 mg/dL, showing that African–American ethnicity was more represented in the group with high Lp(a) values, while there were no differences related to gender. In addition, the prevalence of early cardiovascular events in the family was higher in the group with high Lp(a) values, HDL cholesterol values were higher, and triglyceride values were lower. In contrast, there were no differences in carotid intima-media thickness or pulse wave velocity, two surrogate indices of atherosclerosis [74]. A recent study confirmed the absence of significant differences in early vascular aging in a group of young people with elevated Lp(a) compared with a control group [75]. Recently, the Italian LIPIGEN network demonstrated in a group of 653 Caucasian children and adolescents aged 2 to 17 years with clinical and/or genetic diagnosis of familial hypercholesterolemia that individuals with the highest Lp(a) values were also those with a higher prevalence of early cardiovascular events in first- or second-degree gentiles [76]. Finally, a very new publication [77] considered two studies in which the relationship between Lp(a) values in young people aged 9 to 24 years and 8 to 17 years and the incidence of cardiovascular events at the mean age of 47 years was assessed. After adjustment for several other cardiovascular risk factors, Lp(a) values > 30 mg/dL carried about twice the risk of early cardiovascular events. The concurrent presence of elevated LDL cholesterol values doubled this risk. When the data were analyzed continuously, the risk of events increased by 30 percent for every 1 standard deviation increase in Lp(a) value. In children, the clinical diagnosis of FH is based almost exclusively on LDL cholesterol values, since the other clinical signs contributing to the diagnostic scores, which are present in adults, are never found in children. Because when we assay LDL cholesterol, we cannot distinguish the amount of cholesterol carried by LDL (which increases in the presence of FH) from that related to Lp(a), the presence of high Lp(a) values could lead to misdiagnosis of heterozygous FH. De Boer’s group tested this hypothesis by analyzing a sample of subjects less than 18 years of age referred for family history of FH and/or early cardiovascular disease and/or hypercholesterolemia in the family [78]. The authors noted that children with a diagnosis of FH confirmed by genetic analysis had lower Lp(a) values than those with a phenotype suggestive of FH for elevated LDL cholesterol values but with a negative genetic diagnosis. The latter group was also the one with a higher prevalence of familiarity for early cardiovascular events. These data suggest that high Lp(a) values may be a greater risk factor than carrying a genetic mutation causing heterozygous FH [79]. On the sidelines, we can mention that recently, the analysis of data from the LIPIGEN network in adults confirmed that very high Lp(a) levels can explain at least part of the clinical diagnoses of FH in subjects with very high-LDL cholesterol values and negative analysis for causative genes of this disease [80].

In conclusion, Lp(a) values at birth are very low, but they increase rapidly in the first few months of life. It is believed that definitive values are reached around the age of two years, but some data cast doubt on this assumption. There is evidence, however, of the existence of a correlation between Lp(a) values in childhood and in adulthood. There also seems to be a correlation between child and parental Lp(a) values. Finally, high Lp(a) values in children and adolescents correlate with a higher frequency of cardiovascular events in the family and an important increased risk of early cardiovascular events.

## 6. Role of Lp(a) in Defining Pediatric Cardiovascular Risk and Preventive Activity

Cardiovascular disease is the leading cause of death in Europe and the World [81,82] and is also a major cause of disability and health spending [83]. In low-income countries, it is not economically and organizationally feasible to provide adequate care for patients with cardiovascular disease [84,85], while in developed countries, where effective and innovative but expensive treatments are available [86], the economic burden of these diseases can become critical for welfare systems [83]. Even with these considerations in mind, primary prevention should be the key choice of a correct and forward-looking health policy. Cardiovascular disease manifests clinically in adulthood, but the underlying atherosclerotic processes are already evident even in the first decade of life [87,88]. Paradoxically, atherosclerosis could be considered a pediatric disease. Prevention of cardiovascular disease should therefore be started very early. It is well known that unhealthy lifestyles and diet, as well as exposure to environmental pollution, promote the development of atherosclerosis [89,90]. Therefore, all children and adolescents should be offered proper nutrition, adequate physical activity, and the opportunity to live in a healthy environment. However, these generalized interventions on the pediatric population should be complemented by a personalized one to early identify those children who already carry risk factors (hypertension, dyslipidemia, alterations in glucose metabolism, hyperuricemia) with the aim of stratifying the risk profile of each individual, as is already the practice for adults. Recently, it has been proposed to include Lp(a) in the algorithms for calculating the cardiovascular risk of the adult population [91]. Although less precisely than in adults, it may also be possible in children and adolescents to define an estimate of future cardiovascular risk, and this may be useful in planning individualized interventions. The questions to be answered are whether pediatric Lp(a) assay is useful, is cost-effective, should be offered to all children as screening and at what age or whether it should be reserved for particularly high-risk individuals. Currently, there is no agreement among Scientific Societies regarding the indication to perform Lp(a) assay to all individuals, despite the evidence for the correlation between Lp(a) and cardiovascular risk. The European Society of Cardiology and the European Atherosclerosis Society suggest dosing everyone at least once in a lifetime [91], while United States Scientific Societies recommend measuring Lp(a) only in at-risk individuals [92,93,94]. On this basis, the position expressed in 2015 by McNeal not to perform Lp(a) assay routinely in young people and to reserve it only for particular cases, suggesting healthy lifestyles for all children and adolescents [95], is consequential. More recently, Khon and colleagues have instead suggested the possibility of generalized screening in pediatric age, albeit with a number of concerns [96]. Concerns relate to the absence of a specific treatment that can reduce Lp(a) levels, the risk of creating worry in families or inducing excessive dietary restrictions in younger children, the uncertainty of what the risk thresholds of Lp(a) values are in children of different ethnicities, and, finally, the lack of evidence on what course of action to take once children with elevated Lp(a) values become adults. An argument against performing generalized pediatric screening is that since the American Academy of Pediatrics’ indication to perform a lipid profile assay to all 10-year-old children is poorly followed, it would be unnecessary to add an indication to also assay Lp(a) [97]. However, the evidence that would suggest the clinical utility of performing Lp(a) dosing before the age of 20 is plentiful [96] and, certainly, it seems reasonable to perform the assay when the LDL cholesterol value is high, in suspicion of heterozygous FH [98,99,100]. So, there is an open discussion on this issue. In this debate, one point to be clarified is what is meant by the term screening. Should Lp(a) dosing be performed in all children, or should the assay be performed only in children at cardiovascular risk, as a component of the risk profile, in addition to family history, anthropometric parameters, blood pressure measurement, and metabolic profile assay (lipid profile, glucose profile, and serum uric acid)?

In conclusion, in our opinion, including the Lp(a) assay within the panel of pediatric cardiovascular metabolic risk factors could be of clinical utility for several reasons. Although it is not entirely clear whether, after the age of two years, the LPA gene is fully expressed or Lp(a) values continue to rise into adulthood, and children with high Lp(a) values will continue to maintain high values as adults. Lp(a) promotes all steps of atherosclerosis, but it is particularly active in promoting the early stages of atherosclerosis because of its ability to enter the subendothelial layers of arteries. Thus, children with high Lp(a) will develop earlier atherosclerotic lesions and, other risk factors being equal, more rapidly progress through the steps leading to atherosclerotic plaque formation. The clinical manifestations of atherosclerosis belong almost exclusively to adulthood, yet atherosclerotic changes in the vessels progress in each individual with a speed proportional to the presence of his/her risk factors throughout life. Thus, true primary prevention is effective only if it is started from an early age. Knowing each individual’s risk early, including by assessing Lp(a) levels, can help implement earlier and more effective prevention. There are currently no effective treatments available to reduce Lp(a), and when there are, young people will not be the first to be offered them. However, the presence of elevated Lp(a) recommends increased attention in controlling other modifiable risk factors and a stronger indication to lead a healthy life beginning in childhood and adolescence. Finally, the finding of elevated Lp(a) values in a child could be a stimulus for the family, in the presence of other risk factors such as excess weight and/or hypertension, to greater adherence to dietary-behavioral or pharmacological treatments that are proposed and could suggest the need for testing in parents.

A final point to clarify is whether to measure Lp(a) in all children and at which age. The value of Lp(a) is genetically determined and is randomly distributed in the population. If the value in the parents is not known, there are no other parameters to decide which children to measure. Considering that Lp(a) dosing has a low cost, it would be appropriate to measure it once in a lifetime in all children, after two years of age. As mentioned above, it would be appropriate to combine the Lp(a) assay with a comprehensive analysis of the individual’s risk factors. An elevated Lp(a) value increases overall cardiovascular risk, becoming more significant in children with familial dyslipidemia, hypertension, obesity, or glucose metabolism disorders. Since the American Academy of Pediatrics recommends that all children between the ages of 9 and 11 be evaluated for lipid status [97], it may be advisable to add the Lp(a) assay at this time. If high Lp(a) values are found in one or both parents, measuring in the offspring is mandatory. The measurement, in our opinion, can be performed after the age of two, at whatever age an improvement in lifestyle and diet is deemed appropriate and effective.

## 7. Conclusions and Research Perspectives

Lp(a) has very low values at birth because its gene activity is not yet fully expressed, and its levels increase during childhood, but it is unclear when values proper to adults are stably reached and whether or not growth continues into adolescence. Thus, in children and adolescents, a single Lp(a) assay may not be sufficient as is suggested for adulthood. However, it seems clear that a child with high Lp(a) will almost certainly become an adult with high values of this lipoprotein, so a child with high Lp(a) values should be considered an individual with increased future cardiovascular risk. The gender and ethnicity differences known in adults have not been clearly demonstrated in pediatric age. Based on these considerations, studies in unselected pediatric populations would be needed to understand whether different Lp(a) threshold values should be defined according to age, gender, and ethnicity. Finally, the cardiovascular risk in individuals with high Lp(a) values has been defined as equal or even greater to that of heterozygous carriers of familial hypercholesterolemia [79], for whom hypolipidemic drug treatment is planned from a pediatric age [98,99]. Therefore, when drugs, currently in trials [46,101], that can reduce Lp(a) values become available, it will be appropriate to evaluate, even in pediatric age, in which individuals these medications might be useful.

## Figures and Tables

**Figure 1 biomedicines-11-01661-f001:**
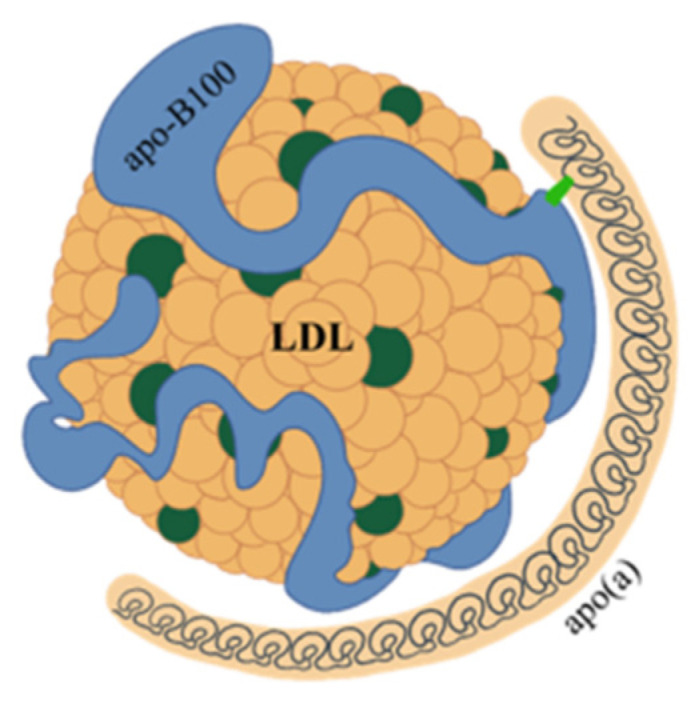
Structure of lipoprotein(a). Lipoprotein(a) consists of a very low-density lipoprotein (LDL)-like particle, with apoprotein B100 (apo-B100) being bound via a single disulfide link to a single apoprotein a (apo(a)).

**Figure 2 biomedicines-11-01661-f002:**
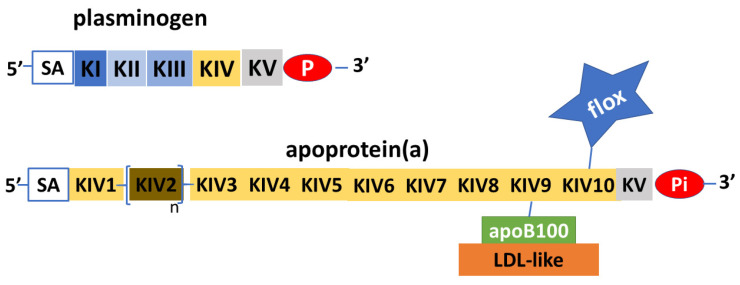
Schematic depiction of plasminogen and apoprotein (a). Apoprotein(a) is a glycoprotein with remarkable similarities to plasminogen, in both the protease part (P) and the inactive part (Pi), and in the kringle protein structure (K). The apoprotein(a) does not have KI to KIII, while it has KIV to KV; KV and subtypes 1 and 3 to 10 are expressed in single copy, while subtype 2 is pre-sent in variable copy numbers. Apoprotein(a) can vary greatly in length and molecular weight. Sialic acid (SA) with other N-glucans and O-glucans has structural functions and protects apoprotein(a) from proteases. Apoprotein(a) transports a considerable amount of oxidized phospholipids (flox), which, added to the LDL-like ones connected to apoprotein(a) via apoprotein B100 (ApoB100), make lipoprotein(a) the major carrier of oxidized phospholipids.

## Data Availability

Not applicable.

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
