# Peer review of "Evidence and Uncertainties on Lipoprotein(a) as a Marker of Cardiovascular Health Risk in Children and Adolescents"

_biomedicines, 2023, doi:10.3390/biomedicines11061661_

Round 1

Reviewer 1 Report

Summary:  Paper is a review that describes lipoprotein(a) or Lp(a) as a risk factor for atherosclerosis.  The review gives a good overview about the change in Lp(a) levels from birth through adulthood and highlights the lack of information available for measuring and treating elevated Lp(a) levels in children and adolescents.

Specific comments:

1) This is an interesting paper that focuses on cardiovascular risk in children and adolescents with elevated Lp(a) levels.  Although it is true that there are no drugs currently approved to specifically lower Lp(a) there are drugs available that have been shown to reduce Lp(a) levels in addition to lowering LDL levels. Niacin has been shown to reduce Lp(a) levels by approximately 20% and PCS9 inhibition has been shown to reduce Lp(a) levels by 25-30%.  It’s also worth noting that statins have been reported to increase Lp(a) levels by 10-20%. This should be mentioned in section 3, Lp(a) and cardiovascular risk.

2) Abstract, line 10-11.  Lp(a) is made up of apo(a) and a LDL-like particle, not a particle of LDL cholesterol.  Please correct this sentence.

3) Abstract, line 37.  Please change “apolipoprotein a” to “apolipoprotein(a)”.

4) Section 2, line 39 (Figure 1).  Please include a brief description of the figure in addition to the abbreviations listed.  This should be done for the other figures as well.

5) Section 2, line 56.  Please change “LDL component” to “LDL-like component”.

6) Section 2, line 57.  Individuals with high Lp(a) values are at greater risk of what?  Please state what the greater risk is.

7) Section 2, line 70.  Please change “The variability of Lp(a)…” to indicate that the variability is in Lp(a) levels.

8) Section 7, line 390.  Please change the sentence “Finally, the risk of cardiovascular risk…” to “Finally, cardiovascular risk…”.

1) Abstract, line 19.  The sentence beginning “The present review makes a revision of the evidence…”.  It may be clearer to say something similar to “The present article reviews the evidence…”

2) Section 2, Figure 2.  Please change “plasminogeno” to “plasminogen”.

3) Section 2, line 59.  Please change “bindings” to “bonds”.  This should also be done on line 63 as well.

4) Section 2, line 76.  The word dosage or dose is used here and in several other places in the manuscript but I think the word “measure” or “measurement “ is what is intended.

Author Response

Please see the responses to the reviewer's comments in the attached file

Reviewer 2 Report

GENERAL COMMENT

This is one of the numerous Review articles published in the last 5 years. Since the topic is hot, and this manuscript focusses on Lp(a) in childhood I am in favor of publishing. Yet there are several points that should be considered by the authors.

SOECIFIC COMMENTS

1.     The authors should explicitly mention a few of the recently published reviews on this topic. In addition, a Special Issue on Lp(a) was published last year by ATHEROSCLEROSIS, and in addition SPRINGER Nature published a whole book on Lp(a) < https://link.springer.com/book/10.1007/978-3-031-24575-6>. Either one might be worth mentioning.

2.     Fig.1 Only kringles are shown for apo(a) and this is actually misleading!

3.     Fig.2: The disulfide bridge between apo(a) and apoB100 is not via KV but rather KIV9.

4.     P.2, line 61: The question of whether the assembly of Lp(a) with apo(a) and apoB-100 takes place in the hepatocyte, on the surface or even outside is not settled yet.

5.     The original literature should be given in Ref. 13.

6.     P.4, line 168: There are numerous reports indication just the opposite, namely that premenopausal women have higher Lp(a) values than man. Please consider this in the manuscript.

7.     P.5, line 205. Ref. 47 from 1991 is not the first publication on this topic.  The first manuscript pblished is from 1991: Strobl W, Widhalm K, Kostner G, Pollak A  Acta Paediatrica Scandinavica. 72(4):505-9, 1983

8.     Last paragraphs from line 345 onwards: There are indications that Lp(a)values rise continuously with age (Pagnan A, Kostner G, Braggion M, Ziron L  Gerontology. 28(6):381-5, 1982 ). Together with the fact that Lp(a) is an acute phase protein and its plasma concentration is significantly increased in kidney patients certainly would  justify that Lp(a) should be measured not only once in life! These facts should be elaborated in the text of the manuscript.

Although the English is understandable, many parts of the manuscrit would gain if they were appropriately adapted.

Author Response

(The authors gave the same response as above.)

Reviewer 3 Report

This review deals with Lp(a) and evidence gaps of its role as a marker of cardiovascular risk in in children and adolescents. The review has a timely topic and it is up-to-date and well-written, in most part. 

It would of interest to explain to the reader on page 2 what is the difference between various KIV subtypes, especially between KIV2 and others.

Page 4 line 144, ref 28 does not give information it is supposed to give. Are the references in wrong order?

Chapter 6: I suggest you discuss also should the lp(a) measurement be performed in a child if the parent has a high lp(a) value, and if yes, at what age.

Minor points:

Page 1 Abstract: “makes a revision”, I would not use the word revision in this instance

Figure 2 has some Italian words in the Figure itself, please use English.

Page 2 line 55: should “mainly related to” be “mainly attached to”?

Page 3, line 76 “when planning the dosage”, should this be “when scheduling the measurement” or similar…?

In various parts, KIV2 is sometimes written with 2 as subscript, sometimes without subscript, please correct.

Page 3 line 222 “however' is”, typo additional apostrophe

Page 4 several instances: odd ratio -> odds ratio

Page 5 line 177 “should be dosed”-> “should be measured”. The same issue with using the word dose/dosing incorrectly is also noted in other parts of the manuscript. Also the word “tracking” seems to be used erroneously in many occasions.

Page 7 line 276 “are not in children.” Some word is missing here.

Page 7 line 324 “Mc Neal” -> McNeal

Ref 13 has a wrong layout

English language needs some correction, see above. 

Author Response

(The authors gave the same response as above.)

Round 2

Reviewer 2 Report

The first version of the manuscript has been amended according to the points raised by this reviewer .

There is, however, one issue remaining related to the sentence “However, from the available data, the decrease of Lp(a) does not appear to be associated with a significant reduction in cardiovascular risk [44,4 5]” lone 180, p. 5. My feeling is that the authors intended to mention that on the basis of current data a significant reduction of CVD risk by lowering Lp(a) awaits further studies. In fact there is one publication published by Jaeger et al. indicating that rigorous reduction of Lp(a) indeed has a positive effect  Jaeger BR, Richter Y, Nagel D, Heigl F, Vogt A, Roeseler E, Parhofer K, Ramlow W, Koch M, Utermann G, Labarrere CA, Seidel D. Longitudinal cohort study on the effectiveness of lipid apheresis treatment to reduce high lipoprotein(a) levels and prevent major adverse coronary events. Nat Clin Pract Cardiovasc Med 2009;6:229–239.

I am not qualified to propose changes

Author Response

There is, however, one issue remaining related to the sentence “However, from the available data, the decrease of Lp(a) does not appear to be associated with a significant reduction in cardiovascular risk [44,4 5]” lone 180, p. 5. My feeling is that the authors intended to mention that on the basis of current data a significant reduction of CVD risk by lowering Lp(a) awaits further studies. In fact there is one publication published by Jaeger et al. indicating that rigorous reduction of Lp(a) indeed has a positive effect  Jaeger BR, Richter Y, Nagel D, Heigl F, Vogt A, Roeseler E, Parhofer K, Ramlow W, Koch M, Utermann G, Labarrere CA, Seidel D. Longitudinal cohort study on the effectiveness of lipid apheresis treatment to reduce high lipoprotein(a) levels and prevent major adverse coronary events. Nat Clin Pract Cardiovasc Med 2009;6:229–239.

We thank the reviewer for his/her comment.
The paragraph has been revised as suggested by the reviewer, and the new reference has been added (new ref 42) and commented on. The numbering of all other references has been updated.